# Depression in Major Neurodegenerative Diseases and Strokes: A Critical Review of Similarities and Differences among Neurological Disorders

**DOI:** 10.3390/brainsci13020318

**Published:** 2023-02-13

**Authors:** Javier Pagonabarraga, Cecilio Álamo, Mar Castellanos, Samuel Díaz, Sagrario Manzano

**Affiliations:** 1Movement Disorders Unit, Neurology Department, Hospital de la Santa Creu i Sant Pau, 08041 Barcelona, Spain; 2Department of Medicine, Autonomous University of Barcelona, 08193 Barcelona, Spain; 3Centro de Investigación en Red sobre Enfermedades Neurodegenerativas (CIBERNED), 28031 Madrid, Spain; 4Department of Biomedical Sciences (Pharmacology), Faculty of Medicine and Health Sciences, University of Alcalá, Alcalá de Henares, 28801 Madrid, Spain; 5Department of Neurology, A Coruña University Hospital and Biomedical Research Institute, 15006 La Coruña, Spain; 6Headaches Unit, Department of Neurology, Hospital Universitario y Politécnico La Fe, 46026 Valencia, Spain; 7Department of Neurology, Infanta Leonor University Hospital, 28031 Madrid, Spain

**Keywords:** depression, dementia, Alzheimer’s disease, Parkinson’s disease, stroke, SSRIs, SNRIs, vortioxetine

## Abstract

Depression and anxiety are highly prevalent in most neurological disorders and can have a major impact on the patient’s disability and quality of life. However, mostly due to the heterogeneity of symptoms and the complexity of the underlying comorbidities, depression can be difficult to diagnose, resulting in limited recognition and in undertreatment. The early detection and treatment of depression simultaneously with the neurological disorder is key to avoiding deterioration and further disability. Although the neurologist should be able to identify and treat depression initially, a neuropsychiatry team should be available for severe cases and those who are unresponsive to treatment. Neurologists should be also aware that in neurodegenerative diseases, such as Alzheimer’s or Parkinson’s, different depression symptoms could develop at different stages of the disease. The treatment options for depression in neurological diseases include drugs, cognitive-behavioral therapy, and somatic interventions, among others, but often, the evidence-based efficacy is limited and the results are highly variable. Here, we review recent research on the diagnosis and treatment of depression in the context of Alzheimer’s disease, Parkinson’s disease, and strokes, with the aim of identifying common approaches and solutions for its initial management by the neurologist.

## 1. Introduction

Neurological diseases represent a significant load of chronic disability and financial burden worldwide, especially as the aging population increases in many countries. The number of people living with dementia increased globally by 117% from 1990 to 2016 [1]. Neurological disorders, including epilepsy, migraines, Alzheimer’s disease (AD), Parkinson’s disease (PD), and strokes, are the third most common cause of disability and premature death in Europe [2]. Psychiatric comorbidities, especially depression and anxiety, are highly prevalent in neurological disorders and impose an even greater impact on both patients and caregivers. Depression has been found to increase disease burden by limiting treatment response, increasing disability, limiting outcomes, reducing the quality of life, and increasing mortality [3,4,5,6].

Despite the high incidence and impact on quality of life, it is commonly accepted that depression in the context of neurological diseases is underrecognized and undertreated. The reasons for this situation are generally attributed to the heterogeneity of the symptoms of depression associated with neurological diseases, and the difficulty of a correct diagnosis in patients who are already affected by cognitive or motor disabilities, or age-related comorbidities. Neuropsychiatry teams are not often available in many healthcare settings, and neurologists often must recognize depressive symptoms associated with neurological conditions and manage depression initially, seeking the assistance of psychiatrists in cases in which depression does not respond to therapy. 

Treatment of depression in neurological diseases includes drugs, cognitive-behavioral therapy (CBT), somatic interventions, or even electroconvulsive therapy, but very often the evidence-based efficacy is extremely limited [7,8,9]. Oral antidepressant therapy is usually the first-line treatment. Many drugs are available, although often their efficacy is controversial depending on the accompanying neurological condition. Other disadvantages to being considered for the pharmacological approach are possible interactions with other medications, and low tolerability due to side effects. No general guidelines for the management of depression in neurological disorders are available, although some studies in the last years offer new insights into the diagnosis and treatment of depression in neurology. In this review, we aimed to bring together the main features of depression in frequent neurological diseases, such as AD, PD, and stroke, with the goal of finding common clinical aspects and treatments. We also highlighted aspects that may help to guide more specific management and treatment responses in different neurological disorders.

## 2. Depression in Neurological Diseases: Common Aspects

The heterogeneity of symptoms of depression makes its diagnosis in the context of neurological disease challenging. Often symptoms of depression can be confused with those of the neurological disease, or with age-related comorbidities. The typical clinical symptoms of depression can be grouped as affective, cognitive, and somatic, and include symptoms such as feelings of worthlessness, anhedonia, depressed mood, guilt, fatigue, difficulty in concentration, suicidal thoughts, and alterations in sleep, weight, and appetite. Depression is usually diagnosed and evaluated mostly based on the presence of affective symptoms only, while other symptoms equally relevant, such as cognitive symptoms, are often overlooked, even if they can be critical for functional recovery and the quality of life of the patient [10,11]. 

Suicidal ideation is a common feature of major depression in patients with neurologic disorders. A significantly higher rate of suicide has been observed in patients with traumatic brain injury, epilepsy, migraine, and multiple sclerosis, as well as in those with degenerative disorders such as AD, Huntington disease, ALS, and PD [12,13,14]. 

If depression is not detected or treated simultaneously with the neurological disorder, this can cause the persistence of symptoms and further disability. The basis for the evaluation of depression in all neurological diseases is the structured interview. To help in the diagnostic process, numerous scales have been developed for the evaluation of depression, although these should not be used in isolation but followed up with a detailed clinical assessment. Some of the most commonly used scales in clinical studies are the Beck Depression Inventory (BDI), the Center for Epidemiologic Studies Depression Scale (CES-D), the Hamilton Depression Rating Scale (HAM-D), the Montgomery-Asberg Depression Rating Scale (MADRS), or the Patient Health Questionnaire-9 (PHQ-9). However, often their use in patients with neurological disorders has not been validated and their use in routine clinical assessments is not standard. It has been suggested that score cut-off adjustments may be needed for some patients in these populations to account for overlapping motor and nonmotor symptoms of depression and to improve their sensitivity [15]. The nine-item and the two-item Patient Health Questionnaire (PHQ-9, PHQ-2) generates reliable results and is easy to administer in clinical settings [16,17]. Its validity in patients with some neurological disorders was recently demonstrated [18], although its reliability and validity in stroke patients were deemed inconclusive in a recent study [16,19]. A major limitation of scales is that they are self-reported and subject to the patient’s insight and subjective judgment, which is often absent in the context of diseases associated with dementia. There is still a need to establish specific depression diagnostic criteria for the different entities that present with dementia, such as AD or PD, because diverse symptoms could develop at different stages of the disease. Therefore, scales likely tend to underestimate the severity of depression. If a caregiver is involved in the completion of the scales for the patient, then the information may not be reliable.

The fact that the prevalence of depression and anxiety among neurology patients is very high suggests shared pathophysiological processes. The etiology of depression in neurological diseases is likely multifactorial and has been attributed to various mechanisms including altered signaling of neurotransmitters, changes in brain structure, inflammation, disrupted neurotrophic factors, and psychosocial agents [5]. A recent study showed that there is a shared genetic basis between AD and depression [20]. In some neurological disorders, such as AD, epilepsy, and stroke, the evidence indicates that there is a bidirectional relationship between these two conditions [5,21]. Thus, depression must be considered a risk factor for certain neurological disorders, and vice versa. However, past episodes of depression or pre-existing depression are associated with an increased risk of depression in patients with vascular dementia, AD, PD, and stroke [22,23,24,25,26]. There is a relationship between the degree of cognitive decline and depressive symptoms over time, suggesting a strong interconnection between depression and degenerative dementias [27,28,29]. However, the exact nature of the relationship between depression and dementia is still a matter of debate [30]. Additionally, in patients with PD other identified risk factors of depression include female sex, early-onset PD, “atypical” parkinsonism, and the presence of psychiatric comorbidities such as psychosis, anxiety, and apathy [25]. For patients with stroke, major predictors of depression are disability, prior cognitive impairment, stroke severity, and anxiety [22,31]. 

Social interactions can be interrupted in patients with neurological disorders as a result of the progression of their disease, leading to social isolation and an increased risk of depression. Depression risk is further increased if the neurological disorder limits the ability of the individual to communicate suicidal thoughts and depressive symptoms, either because of progressive impairment in cognitive abilities or physical problems (e.g., aphasia). For example, numerous recent studies have found an association between hearing loss, depression and cognitive decline [32]. In this context, the COVID-19 pandemic increased isolation and worsened previous neurologic and mood disorders in many patients, and also both neurologic and mood disorders have increased in post-COVID-19 patients [33,34]. The pandemic has had a negative impact on various populations of patients with neurologic disorders [35]. A recent study of patients in Taiwan showed that the COVID-19 pandemic increased symptoms of anxiety and depression among post-acute patients with stroke [36]. However, other studies have shown that anxiety, but not post-stroke depression (PSD), had increased during the pandemic [37,38].

## 3. Overview of Depression in Alzheimer’s Disease

Neuropsychiatric symptoms affect nearly all patients with AD. Specifically for depression, a recent meta-analysis showed that its prevalence in patients with mild cognitive impairment was 25% if derived from community-based samples, and 40% from clinic-based samples [39]. A meta-analysis of the prevalence of major depressive disorders among older adults with dementia of different origins (all-cause dementia), found that its prevalence in patients with AD was 14.8%, and higher (24.7%) in patients with vascular dementia [40]. In addition to AD, depression is also highly prevalent in other dementias, such as Lewy body dementia [41], frontotemporal dementia [42], and Huntington disease [43,44]. If studied attending to the dementia stage, a recent meta-analysis showed that, despite a lack of consensus on the best diagnostic strategy, the progression of AD is associated with more severe apathy and less severe depression and anxiety. Nonetheless, the high prevalence of affective symptoms across dementia stages indicates that affective disorders may interact with each other and interfere with well-being from the earliest stages of the disease [45].

The National Institute of Mental Health (NIMH) criteria of 2002 have been the standard for the diagnosis of depression in AD [46]. The NIMH criteria are simple and based on the presence of three or more symptoms during the same 2-week period and represent a change from previous functioning. At least one of the symptoms must either be (1) depressed mood or (2) decreased positive affect or pleasure [46]. Recently, a validation of these criteria was published, with recommendations for improvements in some aspects. In particular, there is a need for future studies examining the neurobiological substrates of depression diagnosed using the NIMH criteria, and how to measure depression severity to facilitate treatment selection. More studies are required to improve the quality of the evidence that substantiates the items included in the criteria. [47].

As with all neurological diseases, a structured clinical interview is essential during the diagnosis of depression in AD. This can be helped by published instruments and scales specific to patients with AD, but their true validity remains uncertain [8]. Recently, the Eight-item Informant Interview to Differentiate Aging and Dementia (AD8^®^) was found to have high sensitivity in the identification of adults who may benefit from further specialist assessment, in a variety of settings [48]. The Cornell Scale for Diagnosis of Dementia (CSDD) has been recommended in a recent consensus document as useful for screening patients in everyday practice and to detect and assess the severity of depressive symptoms in older dementia patients [49]. Some differences in the manifestations of depression in vascular dementia, PD, or dementia with Lewy bodies have been described [50]. For example, a study showed that pervasive anhedonia had the greatest value for the differential diagnosis of depression between dementia with Lewy bodies and AD [41]. A recent consensus document concluded that depression in early AD could be characterized by somatic symptoms which could be differentiated from apathy by the presence of sadness, depressive thoughts and early-morning awakening. In later phases of AD, symptoms of depression would include sleep-wake cycle reversal, aggressive behavior, and agitation [49].

Based on the available literature, a two-step approach may be considered. First, the initial diagnosis should be based on a structured interview following the different symptoms listed in the Diagnostic Criteria for Depression of Alzheimer’s Disease [46]. In this first step, the use of the AD8^®^ is useful to help discriminate between signs of normal aging and mild dementia. As a second step, and to quantify and monitor the evolution and response of depressive symptoms to interventional therapies, more specific scales that exclude somatic symptoms (CSDD, Geriatric Depression Scale) should be administered.

## 4. Overview of Depression in Parkinson’s Disease

Neuropsychiatric symptoms are common, diverse, and an integral part of PD in its earliest stages, even before the onset of dementia [51]. Since PD is very often accompanied by clinically significant depression, psychosis, and other complications such as impulse control disorders, anxiety, sleep disorders, and apathy, it has been proposed that PD should be classified as a neuropsychiatric disorder [9,52]. Depression can significantly impact the prognosis, caregiver burden, quality of life of the patient and disease course of PD [25]. In fact, these neuropsychiatric symptoms usually account for more disability, worse quality of life, poorer outcomes (including morbidity and mortality), and greater caregiver burden, than the motor symptoms characteristic of PD. Depressive symptoms are an integral part of PD, but they can develop, especially in the initial stages of the disease, as reactive comorbidity caused by sustained stress. In clinical practice, however, this distinction is not relevant to guide therapy, since the use of pharmacological and non-pharmacological therapies is based on the severity of depressive symptoms.

A systematic review analyzing the onset of neuropsychiatric symptoms in PD patients found that affective disorders, of which depression is predominant, typically predate the onset of the characteristic motor symptoms by an average of 4–6 years [53]. Conversely, depressive symptoms in patients with PD can be a predictor of cognitive decline [54] and of impulse control disorders [55,56]. However, it is commonly accepted that affective disorders, such as depression in PD are under-recognized and undertreated [9,57], possibly because some symptoms of depression may be attributed to PD. A systematic review found that clinically significant depressive symptoms could be present in up to 35% of PD patients, with major depressive disorder (MDD) affecting 17% of PD patients, minor depression 22%, and dysthymia 13% [58]. However, another study found symptoms of depression in 70% of PD patients without dementia, anxiety in 69%, and apathy in 48% [51]. Additionally, these neuropsychiatric symptoms, which are associated with longer duration and higher severity of the disease, could be exhibited predominantly in different clusters of patients [51]. Depressive symptoms often correlate with anxiety disorders, cognitive impairment, and psychosis [59].

The symptoms of depression in patients with PD often include apathy, psychomotor retardation, memory impairment, pessimism, irrationality, and suicidal ideation without suicidal behavior [60,61]. A study of depression symptoms in patients with PD found that negative emotions, apathy, and anhedonia were uniquely correlated with depression in these patients [62]. However, these symptoms could be present in distinct populations of PD patients and could be associated with different degrees of motor impairment [63]. This suggests the need to use appropriate scales for each component during diagnosis. It has been suggested that the clinical differentiation of apathy from the emotional symptoms of depression is especially important in PD, as it can guide treatment approaches [64,65]. Other studies indicate that in PD there is a major overlap of apathy with depression and anxiety [66]. The detection of apathy as a consequence of dysfunction of the nucleus Accumbens, anterior cingulate cortex, and functionally related limbic areas produces a state of reduced emotional resonance and anticipatory anhedonia that can be easily confused with depression. Isolated apathy does not respond well to antidepressants, but can be better treated with dopamine agonists or methylphenidate [64]. In addition, isolated apathy not related to depressive symptoms may herald cognitive decline and dementia [67].

Several rating scales have been proposed to evaluate depression in PD [15,68]. The Geriatric Depression Scale (GDS-15) can help in the diagnosis of depression [57]. In a critical review of depression rating scales in PD, the clinician-rated and widely used Hamilton Depression Rating Scale (HAMD-17), and the self-report GDS-15 were recommended for screening and measuring the severity of depression in PD [68]. Again, the GDS-15 may be a preferred choice due to its brevity and ease of use design for older adults, but the authors considered other scales as valid and reliable instruments to use in PD, including self-rated scales such as the Hospital Anxiety and Depression Scale (HADS-D), the Hamilton Depression Inventory (HDI), the Beck Depression Inventory (BDI), and also the observer-report Montgomery-Åsberg Depression Rating Scale (MADRS). It is worth noting that the MADRS, GDS-15, and HADS-D are scales in which the somatic symptoms of PD that may overlap with depressive symptoms are less present, which may facilitate an accurate diagnosis of depression in PD patients with associated non-motor symptoms [69]. 

As for AD, the diagnosis of PD is also based on a two-step approach. First, DSM criteria for depression may guide the detection of depressive symptoms by following a structured interview, and then the use of more specific depression scales that avoid somatic symptoms, such as the GDS-15, HADS-D and MADRS, help to quantify the severity of symptoms.

Since depressive symptoms are an important contributor to disability, poor quality of life, and mortality, the decision to initiate therapy must be taken early in the progression of the disease [4].

## 5. Overview of Depression in Stroke

As is the case in AD or PD, depression is the most frequent affective disorder after a stroke and has a major impact on post-stroke rehabilitation, quality of life, mortality, and disability [22,70]. Systematic reviews have shown that PSD can occur in approximately 1 in 3 patients with a stroke [71,72]. This prevalence remains stable up to 10 years after the stroke [22]. 

The etiology of PSD is complex and largely unknown. Multiple factors and mechanisms (biological and psychological) could contribute to PSD [73]. Early hypotheses on the etiology of PSD pointed to pre-existing micro- or macrovascular cerebral lesions as the underlying cause of the disease in stroke patients [74]. In this view, some of these cardiovascular lesions could have been present before the stroke developed. However, among vascular risk factors, only hypertension can predict PSD, while other factors, such as diabetes, hyperlipidemia, obesity, and smoking, were not independent predictors of PSD [75]. Alternative hypotheses supported by more recent studies points to changes in neurotransmitter balances, inflammatory processes, neural network disfunctions, the roles of homocysteine or vitamin D, or social psychological aspects [73,76,77,78]. To date, no definite association between lesion location and PSD has been found [79,80]. Some studies have pointed to frontal cortical and subcortical structures playing a role in PSD, but more studies are needed [78].

There are no clearly defined specific symptoms of depression in patients with a stroke. Research suggests that the evaluation of depression in the acute phase of stroke is critical [70]. Affective symptoms described in PSD include decreased emotional reactivity, anhedonia, and social withdrawal. Somatic symptoms include fatigue, constipation, anorexia, sleep–wake rhythm disorders, and decreased libido, while cognitive complaints are accompanied by difficulty concentrating, feelings of hopelessness, guilt, and worthlessness and hallucinations [81]. Patients with early PSD have more somatic symptoms, whereas patients with late PSD have more psychological symptoms. A study of patients with first-ever ischemic strokes found that patients often reported crying and sadness, rather than apathy, upon admission into the stroke unit. In these patients crying soon after the stroke, age < 68 years, and severe disability were predictors of PSD within the first year of the stroke [82]. There are overlapping symptoms of PSD and strokes, which makes diagnosis more challenging and could lead to overdiagnosis. However, the presence of some stroke symptoms, such as aphasia or agnosia, and cognitive impairment, may not allow the correct evaluation of PSD and lead to underdiagnosis. 

Prevention of PSD should be focused on the identification of high-risk patients for PSD, and consider possible adverse events associated with SSRI therapy, such as bone fractures (RR 2.28) and nausea (RR 2.05) [83]. The complexity of PSD mechanisms makes its prevention and treatment challenging. Depression after stroke requires close clinical follow-up, especially those patients at the highest risk, and treatments should engage all the healthcare professionals involved in the treatment of stroke patients [84].

An efficient way to detect PSD is the subsequent administration of the 9-Item and 2-Item Patient Health Questionnaire (PHQ-9, PHQ-2) [16]. The use of PHQ aims to early detect PSD as an essential first step for optimizing both therapeutic management of depression, and recovery of post-stroke sequelae at multiple levels. A prospective study of 171 consecutive patients with stroke measured depression in the 6th to 8th weeks after stroke using the PHQ-9 and PHQ-2 and Composite International Diagnostic Interview. Screening of depression was best achieved with a PHQ-9 score ≥ 10 and a PHQ-2 ≥ 2. Interestingly, in the clinical setting, administering the PHQ-9 only to patients who scored ≥ 2 on the PHQ-2 improved the accuracy of screening depression [16].

## 6. Pharmacological Treatment of Depression in Major Neurodegenerative Diseases and Stroke

### 6.1. Common Therapeutic Approaches

A multidisciplinary approach is essential when treating depression in patients with neurological diseases. Integrated assistance considering the neurological, psychiatric, and psychological (cognitive, premorbid, social, and family) aspects of each patient is key to the success of managing depression in neurological diseases. Therapeutically, integrative strategies combining pharmacological strategies with personalized non-pharmacological interventions, such as educational, mental, and physical health support, are needed when neurological symptoms have also an important impact on daily functionality and quality of life.

In terms of evidence-based therapeutic approaches, there is still a need for well-designed randomized controlled trials to test the safety and efficacy of pharmacological as well as non-pharmacological and prevention programs. For many therapies, the available studies are highly heterogeneous in terms of inclusion/exclusion criteria for the patients (e.g., mixed minor and major depression populations), diagnostic techniques (e.g., clinical, scales), methodology (e.g., statistical methods), sample size, and the time of diagnosis of depression with respect to the neurological disease. A large placebo effect has been observed in many randomized studies of antidepressants in PD [85,86]. This heterogeneity often prevents drawing robust or general conclusions. 

For the treatment of MDD, pharmacological treatment is central, but there are no specific guidelines for the election of the antidepressant to use in specific neurological diseases (AD, PD, PSD). The relative efficacy and acceptability of common antidepressants used in clinical practice were recently reviewed in a meta-analysis [87].

### 6.2. Use of SSRIs, SNRIs, and Multimodal Antidepressant Drugs

SSRIs and SNRIs have been used for many years in the treatment of depression and remain the first line of treatment for depression secondary to neurologic conditions [8,57,88]. Their use is preferred over other agents due to their relatively favorable safety profile and ease of administration. However, the use of pure SSRIs has been associated with the presence of apathy [89] and emotional blunting in diseases where there is an additional deficiency of other neurotransmitters such as noradrenaline or dopamine [90,91]. In those situations, such as in PD, and in elderly populations, the use of SNRIs has been seen to significantly reduce the severity or residual apathetic symptoms (100). 

Despite some studies claiming that the use of SSRI for depression in AD has no clear significant effects [92], the limited evidence for supporting the generalized use of antidepressants in dementia [93] is highly attributable to methodological issues and the few randomized clinical trials performed to date, which limits the clear conclusions of their potential role [94]. Considering all these limitations, placebo-controlled studies in recent years have evidenced the efficacy of escitalopram and sertraline for improving depression in AD, as measured by the Hamilton Depression Rating Scale and the Geriatric Depression Scale. Mirtazapine, an antagonist of serotonin receptors and peripheral adrenergic receptors, used at doses up to 45 mg/day, has also shown efficacy in treating symptoms of depression in AD based on network meta-analysis [95,96,97]. 

More data are available on the usefulness of antidepressants for associated neuropsychiatric symptoms. The use of SSRIs and non-SSRIs has been seen to improve psychomotor agitation when depressive symptoms ameliorate, with the subsequent additional improvement of the burden of care and cognitive function [98]. In addition, different clinical consensus reflecting the use of antidepressants in the real world support SSRIs as a first choice for the pharmacological treatment of depression in patients with dementia [8,49]. 

In PD, SSRIs and SNRIs are also considered first-line pharmacological treatment, although the evidence of efficacy is also weak [86,99,100,101]. A study found that SSRIs were associated with greater apathy in patients with PD, while the use of SNRIs was associated with less apathy [102]. In recent randomized controlled trials, citalopram and venlafaxine have proven efficacy in PD depression, as well as tricyclic antidepressants (TCA), especially nortryptiline (10–50 mg/day), whereas the evidence for paroxetine is conflicting. Published meta-analyses of antidepressants in PD have found a higher efficacy for nortryptiline and desipramine over SSRIs and SNRIs. However, TCAs need to be used cautiously in PD considering their anticholinergic effects and potential cardiotoxicity and should be avoided in patients with cardiac problems and orthostatic hypotension. In PD patients with dementia, TCA may produce confusion and delirium [9,86,101,103].

In patients with stroke, SSRIs are also usually first choice, as their use is associated with a significant reduction in depression severity [83,104,105]. Even if SSRIs are the first choice, their use has been associated potentially with the increased risk of intracerebral hemorrhage and the increased risk of bone fracture [83,84,88,105,106]. Over the past 30 years, randomized controlled trials and metanalyses have shown nortriptyline, fluoxetine, citalopram, sertraline, and trazodone to significantly improve post-stroke depression. SSRIs, as in other conditions, have a better efficacy-safety profile than tricyclic antidepressants [107,108]. 

There is an urgent need, however, for the development and adequate testing of new pharmacotherapies for patients with neurological diseases. In these conditions, the antidepressant of choice would be expected to preserve or even improve cognitive functions, and to have an additional impact on functionally impacting symptoms such as sleep, mobility, and fatigue [109].

In the past decade, pharmacological research has focused on the development of new agents for the treatment of not only depressive symptoms, but also cognitive and functional ones, leading to the development of multimodal antidepressants. Vortioxetine is a recently developed antidepressant with a new mechanism of action that combines inhibition of the serotonin (5-HT) transporter and a strong affinity for several 5-HT receptors [110,111,112,113]. Vortioxetine acts as an antagonist of the serotonin 5-HT3, 5-HT1D, and 5-HT7 receptors, as a partial agonist at 5-HT1B receptors, and as a full agonist of 5-HT1A receptors [111]. The result of these effects is the enhancement of serotonin, noradrenalin, dopamine, acetylcholine, and histamine levels in specific areas of the brain [114]. On the basis of a series of large clinical trials [115], vortioxetine was approved in 2013 by the US Food and Drugs Administration for the treatment of MDD [116] and by the European Medicines Agency for the treatment of major depressive episodes [117]. Current guidelines include vortioxetine among first-line treatments for MDD in some countries [118]. In a meta-analysis (17 studies, N = 3653) reviewing the cognitive effects of antidepressants based on neuropsychological tests, vortioxetine had the largest effects on processing speed, executive control, and cognitive control [119,120,121,122]. In a very recent study vortioxetine effectively improved emotional blunting, overall functioning, motivation and energy, cognitive performance, and depressive symptoms in patients with MDD with partial response to SSRI/SNRI therapy and emotional blunting [123]. Another recent randomized clinical trial comparing vortioxetine with other common antidepressants (escitalopram, paroxetine, bupropion, venlafaxine, and sertraline) found that vortioxetine improved cognition (executive functions, selective and sustained attention, short-term memory, recall and nonverbal reasoning ability) and mood in elderly AD patients with depressive symptoms, compared with all the other antidepressants, and was safe and well tolerated [124]. There is also recent evidence in the form of case reports and case series for the efficacy of vortioxetine specifically for the treatment of depressive episodes associated with PD [125,126,127]. In these studies, some of the symptoms characteristic of PD, such as apathy, cognitive function and sleep disturbances, responded well to vortioxetine, both as first-line and after other SSRIs, and no severe side effects were observed. Similarly, vortioxetine might play an important role in the treatment of depression in stroke survivors, because of its effects in improving cognition on one hand, and because of the lack of interaction on aspirin or warfarin pharmacokinetics or pharmacodynamics and no influence on cardiac parameters [7,128]. However, reported studies are very small, and available data are preliminary, so it must be underlined that more research is needed before the role of vortioxetine in improving depression and neurological comorbidities can be established.

Although vortioxetine, duloxetine, and psychostimulants have evidence of independent, direct, and robust effects on cognitive function in major depression, vortioxetine is the only agent that demonstrated efficacy across multiple cognitive domains associated with functional recovery [10]. Vortioxetine is safe and generally well tolerated in both short- and long-term treatment in MDD patients. Some of the tolerability issues seen with other antidepressants, including sexual dysfunction, insomnia-related events, weight gain and discontinuation symptoms occur with a low incidence, which may represent an advantage for vortioxetine during the long-term treatment which is recommended for patients with MDD [129].

## 7. Non-Pharmacological Treatment of Depression in Major Neurodegenerative Diseases and Stroke

Because of the effects of some drugs on the progression of neurologic disorders, often non-pharmacological interventions are favored in the treatment of depression in these patients. For example, physical activity and aerobic exercise may be effective in the management of neuropsychiatric symptoms in AD patients [130]. Dance-based interventions have been shown to be beneficial to alleviate depression among persons with mild cognitive impairment and dementia [131].

Cognitive behavioral therapy (CBT) can be efficacious in patients with PD and mild depression, improve diverse profiles of depressive symptoms, and could be used preferentially in those patients who do not wish to take antidepressants or when pharmacological treatment is not helping [132,133]. Two systematic reviews and meta-analyses concluded that CBT is effective for the management of anxiety and depression in patients with PD, and strongly recommended it [134,135]. 

Other therapies to consider in the treatment of depression in patients with PD are repetitive transcranial magnetic stimulation and electroconvulsive therapy, especially for those with treatment-resistant depression, although the evidence for the efficacy of these treatments is still limited [86].

In Table 1, we are summarizing the different studies showing the efficacy of pharmacological and non-pharmacological therapies in depression associated with AD, PD, and stroke.

## 8. Conclusions

Depression is a frequent affective disorder across neurological disorders, and is often a determinant of the burden of the disease, quality of life, and mortality. Although in the treatment of depression, a multidisciplinary approach is optimal, the neurologist should recognize symptoms, carry on a diagnosis, and start therapy as early as possible. There should be an increased awareness of lesser-known symptoms of depression, such as cognitive decline, which should be evaluated as well. The incidence of suicide is especially high among patients with neurological disorders.

Novel treatments are emerging for the treatment of depression in the context of neurological disorders. Although the efficacy of antidepressants is well consolidated for major depressive disorder, evidence for their role in milder forms of depression often associated with AD, PD, or strokes is weaker. In this regard, it is essential that randomized clinical trials are designed with well-defined criteria and standardized outcome measures that are clinically meaningful for both patients and caregivers. Although pharmacological treatments will continue to be essential in cases of moderate and severe depression [87], non-pharmacologic interventions also deserve a central role in the care of these patients.

## Figures and Tables

**Table 1 brainsci-13-00318-t001:** Pharmacological and non-pharmacological treatment of depression in AD, PD, and stroke.

	AD	PD	Stroke
Pharmacological treatment	Escitalopram 5–20 mg/day (Mean dose = 11.1 ± 3.7 mg/day) RDBPCT; 6 months. HAM-D = −11.1 ± 4.9 points. (*p* < 0.0001). [95]	Citalopram 20 mg/day RDBPCT; 1 month. MADRS = −14 points (*p* = 0.03) [136]	Fluoxetine 20 mg/day RDBPCT; 6 weeks. MADRS = −16.6 points (*p* = 0.02) Additional benefit on motor recovery [137]
	Sertraline 100 mg/day RDBPCT; 3 months. HAM-D = −16.2 points (*p* < 0.001). [96]	Venlafaxine 75–225 mg/day RDBPCT; 3 months. HAM-D = −11 points (*p* = 0.02) [138]	Citalopram 10–40 mg/day RDBPCT; 6 weeks. HAM-D = −9.5 ± 5.6 points (*p* < 0.005) [139]
	Mirtazapine 30–45 mg/day RDBPCT; 13 weeks. CSDD = −5.0 ± 4.9 points. (*p* <0.01). [97]	Nortryptiline 50 mg/day RDBPCT; 2 months. HAM-D = −10.3 points (*p* < 0.002) [140]	-
	Vortioxetine 15 mg/day RDBPCT; 12 months. HAM-D = −7.4 points (*p* < 0.001). Additional benefit on global cognitive function (MMSE) [124]		
Non- pharmacological treatment	Positive effect of aerobic exercise (3–5 times a week) on depression and other neuropsychiatric symptoms. Randomized Controlled Trials. Systematic review; PRISMA guidelines. [130]	Significant improvement of Cognitive Behavioral Therapy on Depression (mean diff = −0.83; *p* < 0.001). RCT compared with clinical monitoring; 12 weeks [132,133,134,135]	

RDBPCT: Randomized Double-Blind Placebo-Controlled Trial; HAM-D: Hamilton Rating Scale for Depression; RSBCT: randomized, single-blind, controlled trial; CSDD: Cornell scale for depression in dementia; MADRS: Montgomery Asberg Depression Rating Scale.

## Data Availability

Data in this review come from all the articles included in References, which can be found in PubMed Central^®^ (PMC), a free full-text archive of biomedical and life sciences journal literature.

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
