# Peer review of "Depression in Major Neurodegenerative Diseases and Strokes: A Critical Review of Similarities and Differences among Neurological Disorders"

_brainsci, 2023, doi:10.3390/brainsci13020318_

Round 1

Reviewer 1 Report

I like the content of this paper. I think it’s pretty useful and will be beneficial for the future readers. I would like to see authors to revise the paper to address the following issues:

1. Improving the overall structure of the paper. As a review, the content seems to a little bit here and there and hard for the first time reader to follow. For example, when author discuss "depression in different diseases" (section 3-6, Depression in AD, depression in PD, depression in stroke), it will be much clearer for readers if all 3 sections follow similar structure, such as overview of the depression in certain disease pharmacological treatment  non- pharmacological treatment  conclusion. The structure above is just a simple example, of course the authors can come up with a more thorough structure.  My main point is, I feel like it’s better to have similar structure set up for each section so readers can easily follow. 

2. for section 6, it will be really nice if the author can organize the content and put their suggestions/guidelines for future researchers/practitioners into a table, with the treatment summary and citation information included.

AD

PD

Stroke

Pharmacological treatment

 treatment1 [citation]

 treatment2[citation]

non- pharmacological

treatment 

 treatment3 [citation]

...

3. in line 138, if it is the first time PSD appear in the paper, it’s recommended to spell the full name rather than acronym.

Author Response

  1. We agree with the reviewer that structuring better the text helps to follow the content of each section. We have changed the titles of each subsection with the intention to clarify to the readers the information provided in each section. The new structure indicates now more clearly that we are initially describing the specific features of depression in each disease, we are then reviewing pharmacological treatment, and we are finally describing studies on non-pharmacological treatments.
  2. We have also followed this suggestion, and we have organized the treatment summary and citation information in a table.

Table 1: Pharmacological and non-pharmacological treatment of depression in AD, PD, and stroke

AD

PD

Stroke

Pharmacological treatment

Escitalopram 5-20 mg/day (Mean dose = 11.1 ± 3.7 mg/day)

RDBPCT; 6 months.

HAM-D = −11.1 ± 4.9 points. (p <0.0001). [95]

Citalopram 20 mg/day

RDBPCT; 1 month.

MADRS = -14 points (p=0.03) [136]

Fluoxetine 20 mg/day

RDBPCT; 6 weeks.

MADRS = -16.6 points (p=0.02)

Additional benefit on motor recovery [107]

Sertraline 100 mg / day

RDBPCT; 3 months.

HAM-D = -16.2 points (p p <0.001). [96]

Venlafaxine 75–225 mg/day

RDBPCT; 3 months.

HAM-D = -11 points (p=0.02) [137]

Citalopram 10-40 mg/day

RDBPCT; 6 weeks.

HAM-D = -9.5 ± 5.6 points (p<0.005) [107]

Mirtazapine 30-45 mg/day

RDBPCT; 13 weeks.

CSDD = −5.0 ± 4.9 points. (p <0.01). [97]

Nortryptiline 50 mg/day

RDBPCT; 2 months.

HAM-D = -10.3 points (p<0.002) [138]

-

Vortioxetine 15 mg/day

RDBPCT; 12 months.

HAM-D = -7.4 points (p<0.001).

Additional benefit on global cognitive function (MMSE)

[124]

Non- pharmacological

treatment 

Positive effect of aerobic exercise (3-5 times a week) on depression and other neuropsychiatric symptoms.

Randomized Controlled Trials. Systematic review; PRISMA guidelines. [130]

Significant improvement of Cognitive Behavioral Therapy on Depression (mean diff = -0.83; p<0.001).

RCT compared with clinical monitoring; 12 weeks [132-135]

RDBPCT: Randomized Double-Blind Placebo-Controlled Trial; HAM-D: Hamilton Rating Scale for Depression; RSBCT: randomized, single-blind, controlled trial; CSDD: Cornell scale for depression in dementia; MADRS: Montgomery Asberg Depression Rating Scale

3. In line 138, we have now spelled the full name for PSD (post-stroke depression) rather than the acronym.

Reviewer 2 Report

Thanks to the author for conducting the review. Although it has summarised some studies, it needs to be more comprehensive and focus on a particular problem. For example, the abstract shows some descriptive information that is primarily not supported by the findings from the review. The authors have reviewed as much as possible without any time frame or evidence shown in a Table to comprehend. Many statements are not supported by references and sometimes not logically followed according to the objective. 

I recommend doing some more work, presenting systematically and comprehensively, possibly with a narrowing objective and submit in the future for publication. 

Author Response

According to the comments of reviewer 2 we have now reviewed in more detail the different studies included in the review, we have summarized them in a table, and we have added new references.

Table 1: Pharmacological and non-pharmacological treatment of depression in AD, PD, and stroke

AD

PD

Stroke

Pharmacological treatment

Escitalopram 5-20 mg/day (Mean dose = 11.1 ± 3.7 mg/day)

RDBPCT; 6 months.

HAM-D = −11.1 ± 4.9 points. (p <0.0001). [95]

Citalopram 20 mg/day

RDBPCT; 1 month.

MADRS = -14 points (p=0.03) [136]

Fluoxetine 20 mg/day

RDBPCT; 6 weeks.

MADRS = -16.6 points (p=0.02)

Additional benefit on motor recovery [139]

Sertraline 100 mg / day

RDBPCT; 3 months.

HAM-D = -16.2 points (p p <0.001). [96]

Venlafaxine 75–225 mg/day

RDBPCT; 3 months.

HAM-D = -11 points (p=0.02) [137]

Citalopram 10-40 mg/day

RDBPCT; 6 weeks.

HAM-D = -9.5 ± 5.6 points (p<0.005) [140]

Mirtazapine 30-45 mg/day

RDBPCT; 13 weeks.

CSDD = −5.0 ± 4.9 points. (p <0.01). [97]

Nortryptiline 50 mg/day

RDBPCT; 2 months.

HAM-D = -10.3 points (p<0.002) [138]

-

Vortioxetine 15 mg/day

RDBPCT; 12 months.

HAM-D = -7.4 points (p<0.001).

Additional benefit on global cognitive function (MMSE)

[124]

Non- pharmacological

treatment 

Positive effect of aerobic exercise (3-5 times a week) on depression and other neuropsychiatric symptoms.

Randomized Controlled Trials. Systematic review; PRISMA guidelines. [130]

Significant improvement of Cognitive Behavioral Therapy on Depression (mean diff = -0.83; p<0.001).

RCT compared with clinical monitoring; 12 weeks [132-135]

RDBPCT: Randomized Double-Blind Placebo-Controlled Trial; HAM-D: Hamilton Rating Scale for Depression; RSBCT: randomized, single-blind, controlled trial; CSDD: Cornell scale for depression in dementia; MADRS: Montgomery Asberg Depression Rating Scale

Reviewer 3 Report

The manuscript Depression in major neurodegenerative diseases and stroke: a critical review of similarities and differences among neurological disorders by Pagonabarraga et al is a review on depression comorbidity with Alzheimer’s disease, Parkinson’s disease, and stroke.

The review is interesting and well-written. There are a number of issues to be addressed by the authors.

Line 114: although age is an important in depression as risk factors differ in age, I would not define it as the most important risk factor. The authors are invited to add references supporting this statement or modify it.

Lines 144-145: the fact that the meta-analysis investigated dementia in general should be added to increase clarity, since the section is focused on AD.

Section in lines 153-175: although many tools are described, the authors should comment/give their opinion about the best approach to be adopted, based on available literature.

Line 159: if proposed improvements are relevant, they should me mentioned.

Line 168: since the section is focused on AD, the reference to “these diseases” should be clarified.

Section 4: the author should expand on whether depressive symptoms are part of PD or a frequent comorbidity and whether this distinction is relevant to guide therapy.

Lines 217-229: as for AD, a comment on best approach would be useful.

Line 237: if findings in ref 71 are included in ref 72, then ref 71 can be omitted.

Lines 167-268: sentence does not make sense, possibly words are missing.

Lines 273-281: are these the only validated tools for diagnosis in PSD or are those that are recommended by the authors?

Line364-365: the sentence should be reformulated as follows: “...guidelines include vortioxetine among first line treatments for MDD...”.

Lines 371-384: since the reported studies are very small and available data are very preliminary, the authors are invited to expand on these limitations for all cited studies, in order to ensure that the readers are aware that much more research is needed before the role of vortioxetine in these comorbidities will be established.

Author Response

We want to thank the reviewer for all the detailed and thoughtful remarks that have helped to improve the manuscript

  1. Line 114: although age is an important in depression as risk factors differ in age, I would not define it as the most important risk factor. The authors are invited to add references supporting this statement or modify it.

We agree with the reviewer, and we have now removed from the text the sentence: “Age is the most important risk factor for both depression and neurodegenerative diseases.”

  1. Lines 144-145: the fact that the meta-analysis investigated dementia in general should be added to increase clarity, since the section is focused on AD.

We have now clarified this point, and we have indicated clearly that the meta-analysis investigated dementia in general.

  1. Section in lines 153-175: although many tools are described, the authors should comment/give their opinion about the best approach to be adopted, based on available literature.

We have now added a sentence giving our opinion about the preferred approach for the diagnosis of depression in AD: “Based on available literature, a two-step approach may be considered. First, initial diagnosis should be based on a structured interview following the different symptoms listed in the Diagnostic Criteria for Depression of Alzheimer Disease [46]. In this first step, the use of the AD8® is useful to help discriminate between signs of normal aging and mild dementia. As a second step, and to quantify and monitor the evolution and response of depressive symptoms to interventional therapies, more specific scales that exclude somatic symptoms (CSDD, Geriatric Depression Scale) should be administered.”.

  1. Line 159: if proposed improvements are relevant, they should me mentioned.

We have now added a sentence on the improvements proposed: “In particular, there is a need for future studies examining the neurobiological substrates of depression diagnosed using the NIMH criteria, and how to measure depression severity to facilitate treatment selection. Also, more studies are required to improve the quality of the evidence that substantiates the items included in the criteria.”

  1. Line 168: since the section is focused on AD, the reference to “these diseases” should be clarified.

It is now clarified in the text.

  1. Section 4: the author should expand on whether depressive symptoms are part of PD or a frequent comorbidity and whether this distinction is relevant to guide therapy.

We have now included an additional sentence in section 4 indicating that “Depressive symptoms are an integral part of PD, but they can develop, especially in the initial stages of the disease, as a reactive comorbidity caused by sustained stress. In clinical practice, however, this this distinction is not relevant to guide therapy, since the use of pharmacological and non-pharmacological therapies is based on the severity of depressive symptoms.” Refs 59, 60.

  1. Lines 217-229: as for AD, a comment on best approach would be useful.

It has now been included as well, with a similar two-step approach: “As for AD, diagnosis of PD is also based on a two-step approach. First, DSM criteria for depression may guide the detection of depressive symptoms by following a structured interview, and then the use of more specific depression scales that avoid somatic symptoms -such as the GDS-15, HADs-D and MADRS- help to quantify the severity of symptoms.”

  1. Line 237: if findings in ref 71 are included in ref 72, then ref 71 can be omitted.

Both references complement each other. We think that the data coming from the longitudinal study in Ref.70 help to understand the epidemiological data coming from the review of Ref. 72.

  1. Lines 267-268: sentence does not make sense, possibly words are missing.

The reviewer is right. The correct sentence is: “Prevention of PSD should be focused in the identification of high-risk patients for PSD, and consider possible adverse events associated with SSRI therapy, such as bone fractures (RR 2.28) and nausea (RR 2.05) [83].”

  1. Lines 273-281: are these the only validated tools for diagnosis in PSD or are those that are recommended by the authors?

In the case of post-stroke depression (PSD) this is the most accepted procedure for diagnosing PSD, and the tools with the optimal validation.

  1. Line364-365: the sentence should be reformulated as follows: “...guidelines include vortioxetine among first line treatments for MDD...”.

Thank you so much for the observation. We have reformulated the sentence as suggested.

  1. Lines 371-384: since the reported studies are very small and available data are very preliminary, the authors are invited to expand on these limitations for all cited studies, in order to ensure that the readers are aware that much more research is needed before the role of vortioxetine in these comorbidities will be established.

We agree again completely with the reviewer. We have explicitly stated these limitations in the text: “However, reported studies are very small, and available data are preliminary, so it must be underlined that more research is needed before the role of vortioxetine in improving depression and neurological comorbidities can be established.

Round 2

Reviewer 1 Report

issues are well addressed, no further comments. 

Reviewer 2 Report

Congratulations!!

Reviewer 3 Report

All issues were adequately addressed.